# From Wrong To Right: A Recursive Approach Towards Vision-Language Explanation

**Jiaxin Ge**[1,2*] and **Sanjay Subramanian** [1] and **Trevor Darrell**[1†] and **Boyi Li**[1†]

[1] UC Berkeley, CA, USA
[2] Peking University, Beijing, China

{gejiaxin, sanjayss, trevordarrell, boyili}@berkeley.edu

## Abstract

Addressing the challenge of adapting pre-trained vision-language models for generating insightful explanations for visual reasoning tasks with limited annotations, we present ReVisE: a **Re**cursive **Vis**ual **E**xplanation algorithm. Our method iteratively computes visual features (conditioned on the text input), an answer, and an explanation, to improve the explanation quality step by step until the answer converges. We find that this multi-step approach guides the model to correct its own answers and outperforms single-step explanation generation. Furthermore, explanations generated by ReVisE also serve as valuable annotations for few-shot self-training. Our approach outperforms previous methods while utilizing merely 5% of the human-annotated explanations across 10 metrics, demonstrating up to a 4.2 and 1.3 increase in BLEU-1 score on the VCR and VQA-X datasets, underscoring the efficacy and data-efficiency of our method.

## 1 Introduction

Explanations for visual reasoning are important in real-world applications (Anderson et al., 2018; Hendricks et al., 2016) like assistive technologies (Dognin et al., 2020) and interactive learning (Misra et al., 2018), but collecting human annotations for these explanations is expensive. The use of language models (LMs) and pre-trained vision-language models (VLMs) have shown promise in explanation generation (Sammani et al., 2022; Plüster et al., 2022). However, generating high quality explanations remains a considerable challenge when annotations are scarce (Bayoudh et al., 2021; Suzuki and Matsuo, 2022).

Previous work has aimed to ameliorate this issue by focusing on enhancing model architecture and subsequent finetuning using large amounts of human-annotated explanations (Sammani et al., 2022; Plüster et al., 2022). Nonetheless, such techniques, reliant on extensive fine-tuning, fall short in the face of limited annotations. Thus, we propose an approach to amplify the model's own reasoning capabilities during inference to generate high-quality explanations. Recent research has demonstrated the efficacy of step-by-step reasoning in language and multimodal reasoning, particularly in contexts where samples are limited (Wei et al., 2022b; Lu et al., 2022; Zhang et al., 2023; Ge et al., 2023). As such, we adopt a phased approach, integrating visual and linguistic components for step-by-step vision-language explanation.

In this work, we introduce the **Re**cursive **Vis**ual **E**xplanation (ReVisE) — a method for generating visual reasoning explanations that surpasses previous methods while using merely 5% of the human-annotated explanations. Initially, we fine-tune BLIP-v2 (Li et al., 2023) to generate explanations on 5% of the dataset. During inference, we generate an initial explanation, then iteratively generate new explanations based on the preceding one. Each step involves computing new visual features, guided by the preceding sentence. This sentence and the new visual features then serve as inputs to generate a new sentence in the next step. Crucially, ReVisE serves as a dynamic, self-correcting mechanism by progressively redirecting visual attention on the image and regenerating the explanation over steps. Additionally, ReVisE generates pseudo-ground truth explanations for few-shot self-training, producing pseudo-labels that considerably aid self-improvement compared to traditional pseudo-labels.

We evaluate ReVisE on four vision-language natural language explanation (VL-NLE) tasks — e-SNLI-VE (Do et al., 2020), VQA-X (Park et al., 2018), AOK-VQA (Schwenk et al., 2022), and VCR (Zellers et al., 2019). Our results show improvements across ten evaluation metrics,

---

*Work done while visiting UC Berkeley; Code Available at https://github.com/para-lost/ReVisE

†Equal advising.

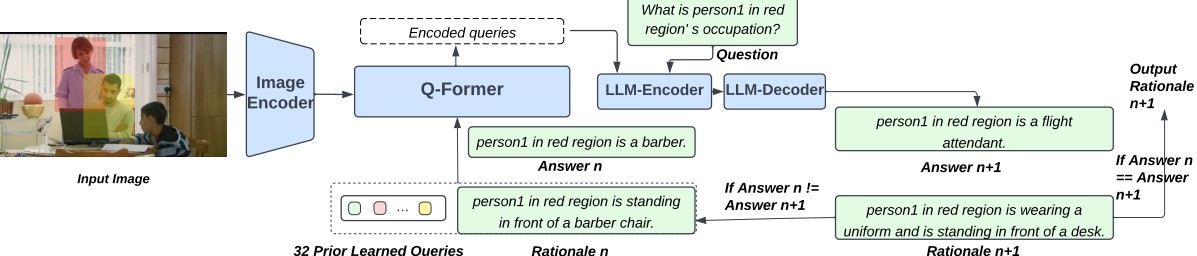

Figure 1: The pipeline of ReVisE. In each step, QFormer receives the concatenated input, consisting of $K = 32$ pre-trained queries and the explanation generated from the previous step, to calculate cross-attention with the encoded image. The output from QFormer, processed further, pairs with the question to guide the frozen LLM in generating the next-step explanation.

with enhancements of up to 4.2 and 1.3 in the BLEU-1 score on VCR and VQA-X respectively. Furthermore, self-training using our method's pseudo-ground truth explanations shows consistent progress compared with traditional generation-based self-training.

Further in-depth ablation studies elucidate the impact of ReVisE and the insights behind it, indicating that sentences can effectively guide visual attention, and that sentence structure and phrasing style are pivotal for few-shot self-training.

Our contribution are summarized as follows:

- We demonstrate that recent pre-trained models require only a fraction of the annotations used by previous explanation generation approaches to reach the same quality.

- We proposed and implemented the Recursive Recursive Visual Explanation (ReVisE), a method that iteratively refines explanations by re-computing visual features.

- We show that self-training using ReVisE to produce pseudo-ground truth annotations further improves the quality of explanations.

## 2 Related Work

**Vision-Language Models (VLM)** Large Vision-Language models have showcased significant potential in vision-language tasks, including VQA, image captioning, and image-text retrieval (Li et al., 2023; Alayrac et al., 2022; Bao et al., 2021; Chen et al., 2022; Li et al., 2021, 2020; Wang et al., 2021; Kim et al., 2021; Bao et al., 2022). Recently, BLIPv2 (Li et al., 2023) was proposed. This model aligns vision with language through a light-weighted transformer architecture (QFormer), rendering it computationally efficient for training on downstream tasks.

**Vision-Language Natural Language Explanation (VL-NLE)** VL-NLE tasks demand a comprehensive understanding and reasoning across both vision and language modalities (Kim et al., 2021). There are two prevailing strategies: the first is a modular approach integrating two separate modules—one for predicting an answer, and another for generating an explanation—represented by works such as e-UG (Kayser et al., 2021), PJ-X (Park et al., 2018), FME (Wu and Mooney, 2018), RTV (Marasovic et al., 2022), and QA-only (Kayser et al., 2021). The second approach is a unified one that uses a single model to generate an answer and explanation simultaneously; relevant works include NLX-GPT (Sammani et al., 2022) and OFA-X$_{MT}$ (Plüster et al., 2022). Our work utilizes the more efficient and training-effective unified approach. However, these methods fall short in effectively integrating the reasoning alignment between vision and language, a gap we address with ReVisE.

**Vision-Language Reasoning** Vision-language reasoning is a cornerstone of vision-language explanation generation. (Hendricks et al., 2016) spearheaded the field by deriving visual explanations from deep networks. (Anderson et al., 2022) focused on visually-grounded navigation instructions, while (Park et al., 2019) applied temporal reasoning to visual tasks. Recently, chain-of-thought (Wei et al., 2021, 2022b,a) has been harnessed to approach tasks using a step-by-step reasoning methodology, effectively enhancing the coherence and logical flow of language reasoning (Wei et al., 2022b; Wang et al., 2022a), self-consistency (Wang et al., 2022b; Lyu et al., 2023) and multimodal reasoning. (Lu et al., 2022; Zhang et al., 2023; Ge et al., 2023)

**Few-Shot Self Training** Self-training, a technique which uses a trained model to generate pseudo-labels for unlabeled data for further model training, improves the model's robustness (Chen et al., 2020; Hendrycks et al., 2019) and benefits vision-language tasks (Baevski et al., 2022; Zhu et al., 2020; Wu and Mooney, 2019). Few-shot self-training, which trains on a small number of samples with pseudo-labels, is used to enhance model performance when data resources are scarce or training costs are high (Li et al., 2019; Mukherjee and Awadallah, 2020; Chen et al., 2021). However, the quality of self-generated pseudo labels greatly influences the effectiveness of few-shot self-training (Zou et al., 2019; Xie et al., 2020; Li and Zhou, 2005). In this work, we demonstrate that ReVisE can generate robust pseudo-explanations beneficial for few-shot vision-language self-training.

**Iterative computation of visual features** The iterative computation of visual features based on text have been applied in previous works to refine visual grounding of an image (Yang et al., 2020), which showed that re-computing visual feature can benefit the visual attention. However, how re-computing benefits text generation remains unexplored. Our work focuses on the text-generation task and shows that the iterative approach can simultaneously benefit the grounding of an image and the quality of the generated text.

## 3 Method

In this section, we first provide an overview of the architecture of BLIPv2 (Li et al., 2023) and how we trained BLIPv2 for VL-NLE tasks. Then, we provide a detailed introduction and pseudo code for ReVisE. Finally, we discuss how ReVisE is employed for self training.

### 3.1 Finetuning BLIPv2 on VL-NLE

BLIPv2 is a generative vision-language model that provides a powerful tool for bridging the divide between vision and language. Its architecture features a lightweight, multi-layer transformer, the QFormer, which computes cross-attention between $K = 32$ pretrained query tokens and encoded image features. Instead of jointly training a text encoder and a vision encoder as in traditional models, BLIPv2 takes a novel approach by freezing the language model parameters and only train the vision encoder and QFormer to convert image features into tokens interpretable by the language model.

This strategy enhances the integration between visual and language components, leading to better task comprehension.

Given an image denoted as $I$. We use the image encoder $E_{image}$ to encode the image to get the image features $F_I$. We denote $K$ BLIPv2 pretrained tokens as $T$. These tokens, together with $F_I$, are passed to the QFormer $QF$, which processes $F_I$ and $T$ to produce the image queries $Q_I$:

$$Q_I = QF(E_{image}(I), T) \tag{1}$$

We denote the tokenized prompt as $P$, with the format "Answer the question by reasoning step by step. Question: {} Answer:". We concatenate $Q_I$ and $P$ to form the full input $F$ to the language model $L$, then we feed it into the language model and obtain the output generated by language model $O$:

$$O = L(concat(Q_I, P)) \tag{2}$$

We calculate a cross-entropy loss $L_{CE}$ between the generated output $O$ and the ground truth sentence $G$, which is constructed in the format "[answer] because [explanation]". :

$$L_{CE} = -sum(G * log(O)) \tag{3}$$

The model parameters are updated to minimize this loss. Only the parameters of the QFormer and the vision encoder are updated while the parameters of the language model are kept frozen.

### 3.2 Recursive Visual Explanation (ReVisE)

---
**Algorithm 1** Pseudo Code for ReVisE
---
1: **Input:** Image $I$, Question $Q$
2: **Output:** Final Answer $A_n$, Explanation $E_n$
3: $F_I = E_{image}(I)$
4: $n = 0$
5: **while** $A_n \neq A_{n-1}$ **do**
6:     $E_n = Tokenize(A_n)$
7:     $E_{n,embedded} = Embed(E_n)$
8:     $Concat_n = concat(E_{n,embedded}, T)$
9:     $Q_{I,n} = QF(Concat_n, F_I)$
10:     $A_{n+1}, E_{n+1} = L(Q_{I,n})$
11:     $n = n + 1$
12: **end while**
13: **return** $A_n, E_n$

---

Given an image $I$ and question $Q$, we first encode the image into a feature set $F_I$ through the image encoder $F_I = E_{image}(I)$ and obtain initial image queries using the pretrained $K$ queries through the QFormer $Q_I = QF(F_I, T)$. We then initialize our iterative steps indexed by $n = 0$. At

the very first step $n = 0$, we feed the image queries $Q_I$ and question $Q$ into the model to generate an initial answer $A_0$ and explanation $E_0$,

$$A_0, E_0 = L(concat(Q, F_I)) \quad (4)$$

For each following iteration $n > 0$, the output $O_n$ is of the form "[answer] because [explanation]". We tokenize the explanation part of $O_n$, denoted as $E_n$. The tokenized explanation $E_n$ is then fed through an embedding layer.

$$E_{n,embedded} = Embed(E_n). \quad (5)$$

We then concatenate $E_{n,embedded}$ with the $K$ BLIPv2 pretrained tokens $T$ to create $Concat_n = concat(T, E_{n,embedded})$. This concatenated structure $Concat_n$ is then passed into the QFormer to calculate a cross attention with the image feature set $F_I$, which then generates a new image query $Q_{I,n}$ based on the explanation $E_n$, $T$, and $F_I$

$$Q_{I,n} = QF(Concat_n, F_I) \quad (6)$$

This new image query $Q_{I,n}$ is then used as input to the language model $L$, which regenerates an explanation and an answer for the next step $n + 1$, denoted as $A_{n+1}$ and $E_{n+1}$

$$A_{n+1}, E_{n+1} = L(Q_{I,n}) \quad (7)$$

This process is repeated recursively until the model converges in its answer.In practice, we limit the maximum iteration number to 5 to prevent potential non-convergence. We provide a pseudo code in Algorithm 1 and a method pipeline in Figure 1.

### 3.3 ReVisE for Self Training

ReVisE's recursive querying process allows the model to correct its own answers, which could lead to further performance improvement. Leveraging this, we propose a few-shot self-training mechanism using the explanations generated by ReVisE. Suppose we have a set of samples $\mathcal{S}$ for which we have the ground-truth answers but lack annotated explanations. Initially, we randomly select a few-shot subset $\mathcal{S}' \subseteq \mathcal{S}$ such that the model originally incorrectly answers these instances, but corrects its answers through ReVisE. Let $A_i^{corr}$ denote the correct answer and $E_i^{ReVisE}$ the explanation generated by ReVisE for the $i$th sample in $\mathcal{S}'$. We then use these pairs, $(A_i^{corr}, E_i^{ReVisE})$, to further finetune the model. During this phase, we freeze both

the language model and the vision encoder, leaving only the QFormer for further finetuning.

$$\theta_{QF}^{new} = \arg\min_{\theta_{QF}} \sum_{i \in \mathcal{S}'} \mathcal{L}(A_i^{corr}, E_i^{ReVisE}; \theta_{QF}) \quad (8)$$

where $\mathcal{L}$ denotes the loss function, $\theta_{QF}$ represents the parameters of the QFormer, and $\theta_{QF}^{new}$ are the updated parameters. This finetuning procedure is designed to bolster the model's ability to generate accurate and explanatory responses.

We contrast this self-training strategy with a traditional approach. In the traditional approach, the model is given the correct answer directly to generate an explanation $E_i^{gen}$ whereas in our approach $E_i$ is generated through recursive querying. In the traditional self-training approach, the model parameters are updated as follows:

$$\theta_{QF}^{new} = argmin_{\theta_{QF}} \sum_{i \in \mathcal{S}'} \mathcal{L}(A_i^{corr}, E_i^{gen}; \theta_{QF}), \quad (9)$$

By juxtaposing these two self-training strategies, we aim to assess the potential benefits of our proposed method, where explanations generated by ReVisE serve as a corrective mechanism, over the conventional approach that relies solely on the model's ability to self-generate explanations from provided answers. A pseudo code is in Appendix C.

## 4 Experiments

In this section, we first introduce the basic settings including the task formulation, training details, baselines, and metrics. Then, we provide detailed experiment results and in-depth analysis for the results.

### 4.1 Settings

**Task Formulation** Our focus is on Vision-Language Natural Language Explanation (VL-NLE) tasks which demand generating an answer and a high-quality explanation given an image-question pair. We test our method on three established VL-NLE datasets (VQA-X (Park et al., 2018), e-SNLI-VE (Do et al., 2020), and VCR (Zellers et al., 2019)), and provide additional results for AOK-VQA (Schwenk et al., 2022). Appendix E provides detailed dataset descriptions.

**Implementation Details** For finetuning BLIPv2 on VL-NLE tasks, we maintain language model frozen and concurrently fine-tune the vision encoder with the QFormer, adopting a learning rate of $1e-5$. We use the entirety of VQA-X while only selecting

Table 1: Filtered Scores comparison for VCR, e-SNLI-VE, and VQA-X against state-of-the-art models. Our BLIPv2 model is fine-tuned on 5% of the VCR and e-SNLI-VE datasets and on the complete dataset for VQA-X while others are all finetuned on the full dataset.

| | B1 | B2 | B3 | B4 | M | R-L | C | S | BS |
|---|---|---|---|---|---|---|---|---|---|
| | | | | | VCR | | | | |
| PJ-X | 21.8 | 11.0 | 5.9 | 3.4 | 16.4 | 20.5 | 19.0 | 4.5 | 78.4 |
| FME | 23.0 | 12.5 | 7.2 | 4.4 | 17.3 | 22.7 | 27.7 | 24.2 | 79.4 |
| e-UG | 20.7 | 11.6 | 6.9 | 4.3 | 11.8 | 22.5 | 32.7 | 12.6 | 79.0 |
| QA-Only | 18.0 | 10.2 | 6.0 | 3.8 | 11.2 | 22.0 | 30.6 | 11.6 | 78.9 |
| RTV | 18.0 | 10.2 | 6.0 | 3.8 | 11.2 | 21.9 | 30.1 | 11.7 | 78.9 |
| OFA-X$_{MT}$ | 22.3 | 13.0 | 8.0 | 5.2 | 11.3 | 24.3 | 44.6 | 17.8 | 79.3 |
| NLX-GPT | 24.7 | 15.0 | 9.6 | 6.6 | 12.2 | 26.4 | **46.9** | 18.8 | 80.3 |
| **ReVisE (Ours)** | **28.9** | **21.7** | **17.6** | **14.4** | **15.5** | **29.5** | 40.2 | **27.9** | **82.2** |
| | | | | | e-SNLI-VE | | | | |
| PJ-X | 29.4 | 18.0 | 11.3 | 7.3 | 14.7 | 28.6 | 72.5 | 24.3 | 79.1 |
| FME | 30.6 | 19.2 | 12.4 | 8.2 | 15.6 | 29.9 | 83.6 | 26.8 | 79.7 |
| RVT | 29.9 | 19.8 | 13.6 | 9.6 | 18.8 | 27.3 | 81.7 | 32.5 | 81.1 |
| QA-only | 29.8 | 19.7 | 13.5 | 9.5 | 18.7 | 27.0 | 80.4 | 32.1 | 81.1 |
| e-UG | 30.1 | 19.9 | 13.7 | 9.6 | 19.6 | 27.8 | 85.9 | **34.5** | **81.7** |
| OFA-X$_{MT}$ | 32.4 | 21.8 | 15.2 | 10.8 | 17.9 | 31.4 | 108.2 | 32.8 | 80.4 |
| NLX-GPT | 37.0 | 25.3 | 17.9 | 12.9 | 18.8 | 34.2 | 117.4 | 33.6 | 80.8 |
| **ReVisE (Ours)** | **38.3** | **26.5** | **19.0** | **13.8** | **19.7** | **34.7** | **126.7** | 34.2 | 81.5 |
| | | | | | VQA-X | | | | |
| PJ-X | 57.4 | 42.4 | 30.9 | 22.7 | 19.7 | 46.0 | 82.7 | 17.1 | 84.6 |
| FME | 59.1 | 43.4 | 31.7 | 23.1 | 20.4 | 47.1 | 87.0 | 18.4 | 85.2 |
| e-UG | 57.3 | 42.7 | 31.4 | 23.2 | 22.1 | 45.7 | 74.1 | 20.1 | 87.0 |
| QA-Only | 51.0 | 36.4 | 25.3 | 17.3 | 18.6 | 41.9 | 49.9 | 14.9 | 85.3 |
| RTV | 51.9 | 37.0 | 25.6 | 17.4 | 19.2 | 42.1 | 52.5 | 15.8 | 85.7 |
| OFA-X$_{MT}$ | 64.0 | 49.4 | 37.6 | 28.6 | 23.1 | 51.0 | 110.2 | 22.6 | 86.8 |
| NLX-GPT | 64.2 | 49.5 | 37.6 | **28.5** | 23.1 | 51.5 | **110.6** | 22.1 | 86.9 |
| **ReVisE (Ours)** | **64.6** | **50.0** | **37.7** | 28.2 | **23.2** | **51.8** | 108.9 | 22.6 | **88.1** |

a random 5% subset from e-SNLI-VE and VCR and AOK-VQA. Under the few-shot self-training scenario, we use 32 examples and exclusively fine-tune the QFormer, applying a learning rate of $1e-6$. More implementation details are provided in Appendix B.

**Baselines** For finetuned BLIPv2, we compare it with previous state of the art models that uses either unified approach or modular approach on the three VL-NLE datasets, including e-UG (Kayser et al., 2021), PJ-X (Park et al., 2018), FME (Wu and Mooney, 2018), RTV (Marasovic et al., 2022), QA-only (Kayser et al., 2021), NLX-GPT (Sammani et al., 2022), OFA-X$_{MT}$ (Plüster et al., 2022). We provide backbone information in Appendix A.

**Evaluation Metrics** In keeping with established practices, we employ N-gram scores, including BLEU (Papineni et al., 2002), METEOR (Banerjee and Lavie, 2005), ROUGE (Lin, 2004), CIDEr (Vedantam et al., 2015), SPICE (Anderson et al., 2016), and BERTScore (Zhang et al., 2019). We also use a more recent metric, G-Eval (Liu et al., 2023), which uses GPT4 (Bubeck et al., 2023) and Auto-Chain-Of-Thought (Zhang et al., 2022) for evaluation that has been shown to align better with

human evaluations. Details of these metrics are available in Appendix D. In accordance with established methods, we present filtered scores that represent results for explanations accompanied by correct answers. Additionally, we also report scores for instances where incorrect answers were given, providing a comprehensive view of the model's performance.

## 4.2 Finetuned BLIPv2

In Table 1, we present our method's performance against other state-of-the-art models using filtered scores for consistency. Leveraging only 5% of the VCR and e-SNLI-VE datasets and the entire VQA-X dataset, we managed to match or exceed benchmark scores with substantially less data. This highlights that advanced pre-trained models like BLIPv2 can achieve comparable performance using fewer annotations. The unique design of BLIPv2, which preserves the language model while transforming visual features into language model-interpretable tokens, offers a promising avenue for future vision-language model architecture research.

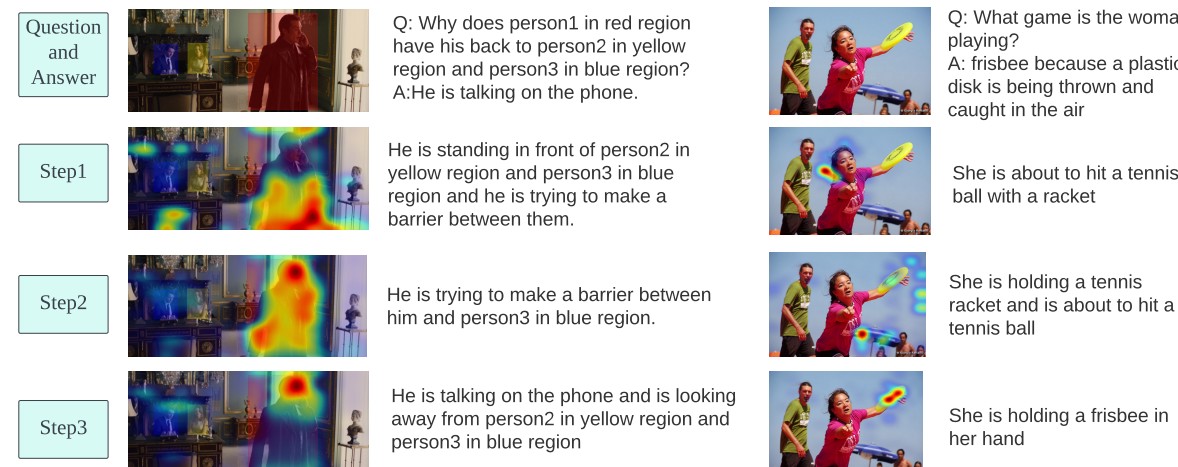

Figure 2: We provide case study of the ReVisE process. We use grad-cam to visualize how the visual attention changes along with how language explanation changes over steps.

Table 2: ReVisE Improvement Scores for VQA-X, eSNLI-VE, AOKVQA, and VCR. Our approach was evaluated on samples initially misinterpreted by the BLIPv2 model. The score of state of the art model NLX-GPT on the same metric is also provided for reference.

|  | B1 | B2 | B3 | B4 | M | R-L | C | S | BS | G-Eval |
|---|---|---|---|---|---|---|---|---|---|---|
| e-SNLI-VE | | | | | | | | | | |
| NLX-GPT | 31.9 | 20.3 | 13.3 | 9.0 | 16.4 | 27.8 | 85.1 | 31.2 | 79.52 | 5.42 |
| Ours (w/o ReVisE) | 35.0 | 22.7 | 15.2 | 10.3 | 17.9 | 29.9 | 101.0 | 30.6 | 79.30 | 6.21 |
| Ours(w/ ReVisE) | **36.7** | **23.7** | **15.8** | **10.8** | **18.2** | **31.1** | **104.0** | **31.4** | **79.90** | **6.64** |
| VQA-X | | | | | | | | | | |
| NLX-GPT | 51.7 | 35.3 | 23.6 | 16.1 | 16.9 | 40.3 | 59.1 | 14.5 | **83.77** | 2.62 |
| Ours(w/o ReVisE) | 51.1 | 34.3 | 22.6 | 14.6 | 15.8 | 39.6 | 51.7 | 12.6 | 83.28 | 2.98 |
| Ours(w/ ReVisE) | **54.8** | **37.3** | **25.0** | **16.2** | **17.6** | **41.3** | **62.6** | **15.0** | 83.52 | **4.24** |
| AOK-VQA | | | | | | | | | | |
| NLX-GPT | 55.1 | 38.3 | 27.1 | 18.1 | 16.2 | 44.0 | 57.4 | 14.3 | 85.43 | 4.12 |
| Ours(w/o ReVisE) | 57.5 | 39.9 | 28.1 | 19.0 | 16.5 | 44.4 | 59.1 | 15.3 | **86.36** | 4.46 |
| Ours(w/ ReVisE) | **59.7** | **41.5** | **28.9** | **19.7** | **17.7** | **44.6** | **60.4** | **16.8** | 85.86 | **4.82** |
| VCR | | | | | | | | | | |
| NLX-GPT | 18.5 | 9.7 | 5.4 | 3.2 | 9.0 | 20.1 | 24.5 | 12.6 | 73.64 | 2.01 |
| Ours(w/o ReVisE) | 26.7 | 19.4 | 15.6 | 12.7 | 13.1 | 24.6 | 19.6 | 21.7 | 79.25 | 3.65 |
| Ours(w/ ReVisE) | **27.2** | **20.3** | **16.4** | **13.4** | **14.1** | **26.2** | **28.7** | **23.7** | **79.35** | **3.97** |

## 4.3 Recursive Visual Explanation (ReVisE)

In Table 2, we showcase ReVisE's impact on augmenting model performance. As our approach aims at self-correcting and refining initially poor-quality explanations, we evaluate ReVisE on samples initially misinterpreted by the BLIPv2 model. The process involves using recursive language querying to extract pertinent image features, progressively refining the model's output. We find that ReVisE persistently enhances the quality of the generated explanations, underscoring the critical role of language as a guide for image feature extraction.

Figure 2 exhibits representative examples from the VL-NLE datasets, clearly demonstrating the self-correcting mechanism of ReVisE. Employing grad-CAM visualizations (Selvaraju et al., 2017), we

elucidate how ReVisE guides the model's attention allocation over steps. While initially, the attention maps are broad or focus on areas irrelevant to the question, ReVisE's language-guided procedure redirects the model's attention towards areas pertinent to the question at hand, suggesting an improvement in the model's interpretability.

By taking the explanation from one iteration and using it as input for the next, the model refines its interpretation and visual attention. Conceptually, it's analogous to a person rephrasing a statement repeatedly to enhance clarity.

## 4.4 Few-Shot Self-Training

In Table3, we show results for few-shot self-training. We use explanations generated by Re-

Table 3: Performance comparison of ReVisE in a few-shot self-training context for e-SNLI-VE, VQA-X, AOKVQA, and VCR. The table depicts results without self-training, with traditional self-training, and with ReVisE self-training. We use 32-shot in all these experiments.

| | B1 | B2 | B3 | B4 | M | R-L | C | S | BS | G-Eval |
|---|---|---|---|---|---|---|---|---|---|---|
| e-SNLI-VE | | | | | | | | | | |
| No Self-train | 35.0 | 22.7 | 15.2 | 10.3 | 17.9 | 29.9 | 101.0 | 30.6 | 79.30 | 6.21 |
| w/o ReVisE | 34.9 | 22.7 | 15.3 | 10.4 | 17.9 | 29.8 | 100.7 | 30.5 | 79.21 | 6.49 |
| w/ReVisE | **36.2** | **23.5** | **15.8** | **10.9** | **18.2** | **30.5** | **103.2** | **30.7** | **79.61** | **6.75** |
| VQA-X | | | | | | | | | | |
| No Self-train | 51.1 | 34.3 | 22.6 | 14.6 | 15.8 | 39.6 | 51.7 | 12.6 | 83.28 | 2.98 |
| w/o ReVisE | 51.2 | 34.1 | 22.4 | 14.3 | 15.9 | 39.6 | 50.6 | 12.6 | 83.00 | 3.21 |
| w/ReVisE | **53.5** | **36.6** | **24.8** | **16.2** | **16.9** | **40.7** | **58.9** | **13.8** | **83.65** | **4.41** |
| AOK-VQA | | | | | | | | | | |
| No Self-train | 57.5 | 39.9 | 28.1 | 19.0 | 16.5 | 44.4 | 59.1 | 15.3 | 86.36 | 4.46 |
| w/o ReVisE | 57.3 | 40.2 | 28.4 | 19.2 | 16.6 | 44.8 | 61.1 | 15.7 | **86.44** | 4.44 |
| w/ReVisE | **60.0** | **41.1** | **28.7** | **19.6** | **18.6** | **45.1** | **62.4** | **18.1** | 85.28 | **4.71** |
| VCR | | | | | | | | | | |
| No Self-train | 26.7 | 19.4 | 15.6 | 12.7 | 13.1 | 24.6 | 19.6 | 21.7 | 79.25 | 3.65 |
| w/o ReVisE | 26.9 | 19.6 | 15.7 | 12.7 | 13.3 | 25.2 | 21.1 | 21.9 | 79.35 | 3.99 |
| w/ReVisE | **27.1** | **20.1** | **16.2** | **13.3** | **13.7** | **25.4** | **21.6** | **23.1** | **79.55** | **4.14** |

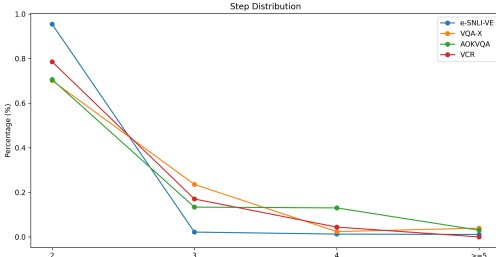

Figure 3: We display the distribution of convergence steps, indicating the percentage of samples that reach convergence at each respective step. We show results of e-SNLI-VE, VQA-X, AOKVQA and VCR and found that most samples converge by step2 and at least 90% samples converge by step3.

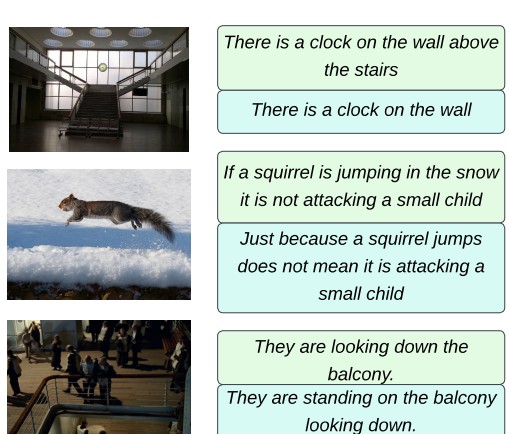

Figure 4: Comparison between pseudo-explanations generated by ReVisE(in the box above) and pseudo-explanations generated directly providing groundtruth answers(in the box below).

VisE to self-train on samples that are initially incorrect but self-corrected during the ReVisE process. When compared to providing the model with the correct answers directly to let it generate an explanation on the same samples, self-training with ReVisE explanations led to better performance, indicating the model benefits more from its own reasoning process than simply digesting the correct answers.

Qualitative results in Figure 4 reveal minor semantic differences between ReVisE-generated pseudo-explanations and explanations crafted with provided ground-truth answers. The variations typically lie in phrasing style or sentence structures, suggesting that attention to sentence structure or phrasing pattern could prove crucial for high-quality pseudo-explanations in few-shot self-training.

Additionally, we explored the impact of varying the number of self-training samples. As shown in Table 5, while any addition of few-shot samples enhances self-training, even as few as 8-shot samples can improve the model's performance.

### 4.5 Ablation Study

**Implicit VS Explicit Language** In our approach, we forward both the integrated $K$ queries and the language queries after cross-attention. We compare this procedure with forwarding only $K$ queries after cross-attention, as illustrated in Figure 6. The $K$ queries integrate language information implicitly through the cross-attention process but do not forward the encoded text directly.

Table 4: Ablation study examining the impact of limiting ReVisE iterations to 2 and 3. Since up to 90% of samples converge by the third step, constraining iteration steps to 3 yields strong results.

|  | B1 | B4 | M | R-L | C | S |
|---|---|---|---|---|---|---|
| e-SNLI-VE |  |  |  |  |  |  |
| 2Steps | 36.6 | 10.6 | 18.1 | 31.0 | 103.0 | 31.0 |
| 3Steps | 36.7 | 10.8 | 18.2 | 31.1 | 103.2 | 31.4 |
| VQA-X |  |  |  |  |  |  |
| 2Steps | 54.6 | 15.7 | 17.5 | 41.0 | 59.6 | 14.9 |
| 3Steps | 54.8 | 16.2 | 17.6 | 41.3 | 60.6 | 15.0 |
| VCR |  |  |  |  |  |  |
| 2Steps | 27.0 | 13.4 | 14.0 | 26.0 | 28.3 | 23.6 |
| 3Steps | 27.2 | 13.4 | 14.1 | 26.2 | 28.7 | 23.7 |

Table 5: Ablation study investigating the effect of varying sample sizes for few-shot self-training. We present results for 8-shot, 16-shot, and 32-shot self-training approaches, using pseudo-explanations generated by ReVisE.

|  | B1 | B4 | M | R-L | C | S |
|---|---|---|---|---|---|---|
| e-SNLI-VE |  |  |  |  |  |  |
| 8-shot | 35.5 | 10.5 | 17.9 | 30.1 | 101.5 | 30.6 |
| 16-shot | 35.7 | 10.6 | 18.0 | 30.2 | 101.8 | 30.5 |
| 32-shot | 36.2 | 10.9 | 18.2 | 30.5 | 103.2 | 30.7 |
| VQA-X |  |  |  |  |  |  |
| 8-shot | 53.3 | 15.8 | 16.7 | 40.6 | 57.4 | 13.2 |
| 16-shot | 53.3 | 15.7 | 16.7 | 40.3 | 56.9 | 13.6 |
| 32-shot | 53.5 | 16.2 | 16.9 | 40.7 | 58.9 | 13.8 |
| VCR |  |  |  |  |  |  |
| 8-shot | 26.9 | 12.9 | 13.3 | 25.2 | 20.7 | 22.3 |
| 16-shot | 27.1 | 13.1 | 13.5 | 25.3 | 21.4 | 22.6 |
| 32-shot | 27.1 | 13.3 | 13.7 | 25.4 | 21.6 | 23.1 |

The ablation study results in Table 6 indicate that implicit language integration through the $K$ queries alone does not significantly enhance performance. Explicitly combining language queries offer crucial semantically-grounded context which cannot be captured by the $K$ learned queries alone, thus providing a more substantial advantage in refining the model's image comprehension.

**Limit Iteration Steps** Recursive querying may be time-consuming, so we limit the maximum number of steps to 2 and 3 to investigate its impact. As shown in Table 4, limiting the steps to 3 achieved performance nearly on par with that of unrestricted steps. Furthermore, Figure 3 presents the percentage of samples that reach convergence at each respective step, indicating that most samples converge by the second step and 90% of the samples converge by step3. The e-SNLI-VE samples exhibit the fastest convergence, potentially due to the simplicity of their answer options.

Table 6: Ablation Study for three VL-NLE datasets. 'I' refers to incorporating language signal implicitly and 'E' refers to incorporating language signal explicitly. Generally, 'E' outperforms 'I'.

|  | B1 | B4 | M | R-L | C | S |
|---|---|---|---|---|---|---|
| e-SNLI-VE |  |  |  |  |  |  |
| I | 35.0 | 10.4 | 17.9 | 29.9 | 100.8 | 30.6 |
| E | **36.7** | **10.8** | **18.2** | **31.1** | **104.0** | **31.4** |
| VQA-X |  |  |  |  |  |  |
| I | 51.0 | 14.8 | 16.0 | 39.9 | 52.4 | 12.5 |
| E | **54.8** | **16.2** | **17.6** | **41.3** | **62.6** | **15.0** |
| VCR |  |  |  |  |  |  |
| I | 24.1 | 11.1 | 12.0 | 24.3 | 18.7 | 19.4 |
| E | **25.5** | **11.6** | **12.4** | **24.5** | **20.0** | **19.8** |

Q:What kind of enemy are person2 in yellow region and the rest up against?
A: A very large and dangerous one.

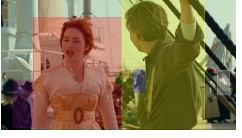

They are wearing sunglasses and holding guns.
**Step1**

They are wearing dark sunglasses and a lot of black clothing.
**Step2**

They are wearing sunglasses and are all wearing suits.
**Step3**

**Failure Case: Explanation Gets Worse Over Steps**

Q: Why is person1 in red region's mouth ajar?
A: person1 in red region is surprised by a joke person2 in yellow region made.

person2 in yellow region is moving to kiss person1 in red region.
**Step1, Step3, ...**

person1 in red region is smiling and person2 in yellow region is laughing.
**Step2, Step4, ...**

**Failure Case: Explanation Never Converges**

Figure 5: Failure cases when iterations doesn't converge or adding more iterations worsens the performance.

**Failure Cases** We notice certain instances ( 2%) where additional iterations negatively affect the quality of the generates explanations, as illustrated in Figure 5. For most failure cases, the model enters into a recursive loop. In some others, the model initially generates explanations closely aligning with the ground truth but diverged with subsequent iterations. This reveals the importance for a balance between the depth of reasoning and model certainty in recursive reasoning.

**Data Efficiency** We provide further ablation on the amount of training data used. On the e-SNLI-VE dataset, we tried 1%, 3%, and 5% of the dataset and report the filtered score. The results are shown in Table 7. This illustrates that our model, leverag-

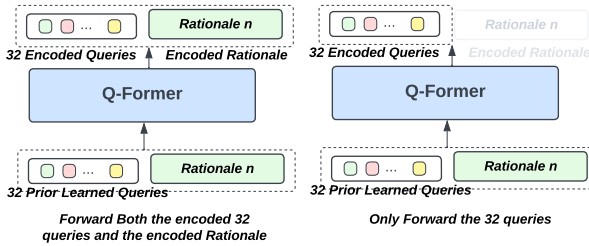

Figure 6: Comparison between forwarding all the encoded queries with forwarding only the $K$ queries after cross-attention.

ing recent advancements in pre-trained models, can deliver high-quality explanations even with substantially fewer annotations than traditional methods.

Table 7: Ablation Study for the amount of data used on e-SNLI-VE dataset. We tried using 1%, 3% and 5% of the training data and reported the filtered scores.

|    | B1   | B4   | M    | R-L  | C     | S    |
|----|------|------|------|------|-------|------|
| 1% | 37.0 | 13.4 | 19.4 | 33.9 | 122.3 | 33.5 |
| 3% | 38.0 | 13.6 | 19.7 | 34.5 | 126.7 | 34.0 |
| 5% | 38.3 | 13.8 | 19.7 | 34.7 | 126.7 | 34.2 |

## 5 Conclusion

In this paper, we introduce ReVisE, a method for generating natural language explanations for visual reasoning tasks with a small number of annotations. We demonstrate its high performance in generating language explanations using significantly less data compared to existing models, enabling the model's self-correction mechanism, and effectively facilitating few-shot self-training by generating robust pseudo-explanations. Our work raises the question of whether recursive procedures like ReVisE can be used to improve performance in other multimodal domains as well.

## Limitations

Althought BLIPv2 has a large potential for vision-language explanation, it might encode social bias. As (Chuang et al., 2023) illustrated, vision-language models have been shown to inherit biases from their training datasets. Conducting a thorough investigation of the potential bias of BLIPv2 and addressing it would be an important future work. Also, further enhancing the method to identify and address the failure cases is also a future work to improve this method.

## Ethics Statement

The proposed methods, rooted in the principles of transparency and interpretability, promote the ethical goal of developing AI systems that are easily comprehensible and verifiable. By enabling AI to generate more coherent explanations, we contribute to the objective of trustworthy AI.

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

## A    Model Backbone

We present model parameters and vision transformer backbone for the different models in Table 8

| Models | Backbone | Trainable Params |
|--------|----------|-----------------|
| FME | ResNet-101 | 142M |
| RVT | ResNet-101 | 277M |
| e-UG | ResNet 101 | 277M |
| OFA-X | ResNet152 | 472M |
| NLX-GPT | ViT | 182M |
| Ours | ViT | 108M |

Table 8: The vision backbone for FME, RVT, e-UG, OFA, NLX-GPT and Ours.

## B    Additional Implementation Details

We provide additional implementation details. When training on BLIPv2 we use beam search with num beam = 5 during decoding. For AOK-VQA, we also set length penalty to -1, consistent with the original BLIPv2. During training, we use cosine annealing sceduler and AdamW optimizer and train for 6 epochs. Since BLIPv2 does not have any regional proposals, we followed (Zellers et al., 2021) and add colored bounding boxes around the people/objects referred to and refer to them as "person1 in red region" or "person2 in yellow region". As (Zellers et al., 2021) demonstrates, through fine-tuning, the model learns a matching between the color referred to in language and the color denoted in the image.

## C    Pseudo Algorithm For ReVisE self-training

We also provide a pseudo code for self-training in algorithm 2, which is a pseudo code description of the self-training process in the Method section.

## D    Detailed Metrics

We provide details of the recent new metric G-Eval (Liu et al., 2023). This metric commences by formulating a task and employs GPT3.5 (Brown et al., 2020) and GPT4 (Bubeck et al., 2023) to autonomously generate evaluation steps using the Auto Chain-of-Thought (AutoCoT) (Zhang et al., 2022). Subsequently, the task instruction along with the AutoCoT evaluation steps and the sample under consideration are fed to the GPT model together to obtain a comprehensive score from 1-10.

**Algorithm 2** ReVisE for Self Training
___

1: **Input:** Model $M$, Samples $S$, Few-shot size $k$
2: **Output:** Finetuned Model $M'$
3: **Initialize:** $TrainingSet \leftarrow \{\}$
4: **for** each $sample \in S$ **do**
5:     $(A_{\text{old}}, E_{\text{old}})$                                    $\leftarrow$
    $M.\text{generateAnswerWithoutReVisE}(sample)$
6:     $(A_{\text{new}}, E_{\text{new}})$                                    $\leftarrow$
    $M.\text{generateAnswerWithReVisE}(sample)$
7:     **if** $M.\text{checkAnswer}(A_{\text{old}})$ is False   and
    $M.\text{checkAnswer}(A_{\text{new}})$ is True **then**
8:         $TrainingSet.\text{add}((sample, E_{\text{generated}}))$
9:     **end if**
10: **end for**
11: Randomly select $k$ samples from $TrainingSet$ to form
    $FewShotSet$ for few-shot self training
12: $M' = M.\text{finetuneQFormer}(FewShotSet)$
13: **return** $M'$
___

This metric has been shown to align better with human evaluations than previous metrics.

## E   Data Details

VQA-X and A-OKVQA both augments the VQAv2 dataset (Antol et al., 2015) with explanations for each answer. The images in VQA-X are sourced from the COCO dataset (Lin et al., 2014), and it comprises 33K QA pairs drawn from 28K images. e-SNLI-VE provides explanations for the Visual Entailment Prediction task, which involves answering whether a given image and hypothesis are in entailment, contradiction, or neutral relationship.The images for this dataset are drawn from Flickr30k (Plummer et al., 2015), and it contains over 430K examples. VCR is a dataset that presents a model with an image, a question, and a list of objects that are annotated with bounding boxes, and requires the model to first select an answer and then explain it. VCR includes 290K samples of questions, answers, and rationales. For each of the dataset, we use the original train set of each dataset and their own test set.