# OpenReview forum: "From Wrong To Right: A Recursive Approach Towards Vision-Language Explanation"
_EMNLP/2023/Conference — EMNLP 2023 Main_

### Official Review · Reviewer_86Pj · 2023-08-03

**Soundness:** 3

**Excitement:**

3: Ambivalent: It has merits (e.g., it reports state-of-the-art results, the idea is nice), but there are key weaknesses (e.g., it describes incremental work), and it can significantly benefit from another round of revision. However, I won't object to accepting it if my co-reviewers champion it.

**Paper Topic And Main Contributions:**

The paper proposes a recursive visual explanation method for step-by-step vision-language explanation generation. The proposed method appears to build upon the powerful BLIP-2 model. Experiments on three datasets demonstrate the effectiveness of the proposed method.

**Reasons To Accept:**

- The paper proposes a multi-step explanation generation method, and the experimental results indicate that the model's performance improves with more steps.
- The paper proposes a few-shot self-training method, enabling training with generated pseudo-explanations.

**Reasons To Reject:**

- Lack of detailed description of the first contribution. 5% of the human-annotated explanations are insufficient to understand the significance of this aspect. The paper should provide more comprehensive information regarding the comparison methods, experimental settings, and detailed analyses to establish a clear contribution.
- It seems that the novelty of the proposed recursive method is relatively incremental, as the iterative computation of visual features conditioned on text input is a common technique in the multimodal domain, as seen in previous work (e.g., Paper [1]). The paper should address how the proposed method extends beyond existing techniques to justify its contribution.
- The paper should include a thorough analysis of the significant performance gap observed between the proposed method and state-of-the-art methods, which are essential for a comprehensive evaluation of the proposed method's effectiveness.

References:
[1] Improving one-stage visual grounding by recursive sub-query construction. ECCV, 2020.

**Reproducibility:**

3: Could reproduce the results with some difficulty. The settings of parameters are underspecified or subjectively determined; the training/evaluation data are not widely available.

**Reviewer Confidence:**

3: Pretty sure, but there's a chance I missed something. Although I have a good feel for this area in general, I did not carefully check the paper's details, e.g., the math, experimental design, or novelty.

---

> ### Author Rebuttal · Authors · 2023-08-28
>
> Thank you for your comprehensive review and insightful comments. We read your comments very carefully and here are some efforts we made that could hopefully address some of your concerns:
>
> 1.**Clarification on the First Contribution and Data Efficiency**
>
> For the first contribution of our experiments, we used **5%** of the training set and tested on the full test set while the other methods are all trained on the full training set and tested on the full test set. We provide further ablation on the amount of training data used. On the e-SNLI-VE dataset, we tried 1%, 3%, and 5% of the dataset and report the filtered score. the results are shown below:
> | | B1         | B2     | B3 | B4 | M | R-L | CIDEr | S |
> |-------------|-------|-------|----------|-----|-----------|---------|-----|-----------|
> | NLX-GPT             |      37.0 |25.3| 17.9 |12.9 |18.8 |34.2| 117.4| 33.6|
> | 1%                    | 37.0 | 25.7| 18.4 | 13.4 | 19.4 | 33.9 | 122.3 | 33.5 |
> | 3%                    | 38.0 | 26.7| 18.9 | 13.6 | 19.7 | 34.5 | 126.7 | 34.0 |
> | 5%                    | 38.3 | 26.5| 19.0 | 13.8 | 19.7 | 34.7 | 126.7 | 34.2 |
>
> Remarkably, using merely 1% of the dataset, our method achieves a performance comparable to the state-of-the-art NLX-GPT. As we scale to 3% and 5% of the data, we observe performance gains beyond the benchmark set by NLX-GPT. This illustrates that our model, leveraging recent advancements in pre-trained models, can deliver high-quality explanations even with substantially fewer annotations than traditional methods.
>
> Additionally, we've elaborated on how both the recent vision-language model BLIPv2 and our proposed method ReVisE contributed to these gains in Table2 and Table3, showing that our improvement gain is caused by both the usage of the recent vision-language model BLIPv2 and also our proposed method ReVisE.
>
>  In addressing the need for a clearer differentiation from existing methods, we've furnished a more detailed breakdown:
> | Method | Vision Backbone         | Language Model     | Training Method |
> |:-------------:|:-------:|:-------:|:----------:|
> | FME                    | ResNet-101 |Up-Down LSTM | Modular |
> | RVT                   | ResNet-101 | GPT-2| Modular |
> | e-UG                    | ResNet-101 | GPT-2| Modular |
> | OFA-X                    | ResNet-152 | OFA[1] | Unified |
> | NLX-GPT                    | ViT | Distilled GPT-2 | Unified |
> | Ours                    | ViT | FlanT5-xxl | Unified |
>
> Where unified model generates answer and explanation together and modular models use two separate modules to first generate the answer and then generate an explanation.
> [1] The text generation architecture of OFA is similar to GPT
>
> Furthermore, we've furnished a detailed experiment setting table for different methods, specifying original settings from each respective paper:
> | Method |  Training Data  | Learning Rate |Optimizer  | Scheduler | Batch Size|
> |:-------------:|:-------:|:-------:|:----------:|:-----:|:-----------:|
> | FMT                    | 100% | 5e-4 | Adam | Step Decay |  128 |
> | RVT                   | 100%|  5e-5 | AdamW | Linear | 32|
> | e-UG                    | 100%| 2e-5 | AdamW | Linear  | 64 |
> | OFA-X                    | 100% | 5e-5 | AdamW | Polynomial |  128 |
> | NLX-GPT                    | 100% | 1e-5 |Adam | Linear |    64 |
> | Ours                    | 5% | 1e-5 | AdamW | Cosine | 2 |
>
> We use a batch size of only 2 due to GPU constraint, we assume that a larger batch size(>=32) would be more beneficial for stable training and potentially further enhance the performance.
>
> 2.**Comparison and Differentiation from Previous Works**
>
> **Task Differences: Text vs. Visual Grounding**:  we focus on the text-generation task which is substantially different from the visual-grounding task seen in earlier works. In our case, we use the step-wise recalculated visual feature and the previous text explanation to generate a new text at each step. In the previous works, the recursive method is used to ground a text to the image region. Even if recursive techniques have been used before, like in the ECCV paper or with RNNs, applying them to our kind of task is not straightforward.
>
> **Simplicity**: Our method is much simpler. We don't require any additional detection models(as in the visual grounding) and we don't need to decompose the question down into sub-queries.
>
> **Dynamic Text Generation**:  One key part of our approach is generating a new text at every step. This is different from just pulling different text features from existing text, which is what existing methods do.

---

### Official Review · Reviewer_yJ8S · 2023-08-05

**Soundness:** 3

**Excitement:**

4: Strong: This paper deepens the understanding of some phenomenon or lowers the barriers to an existing research direction.

**Paper Topic And Main Contributions:**

This paper proposes ReVise, a method that recursively generate explanations for vision-language tasks by recomputing visual features. The method is data efficient and achieves on-par performance with current methods for VL-NLE. They also showed the effectiveness of ReVise in self-training by generating pseudo ground truth annotations for explanations and self-train on these generated explanations.

**Questions For The Authors:**

A. Are the reported metrics evaluated on the answer or the explanation?

B. Have you looked into how explanation quality varies in the iteration process?

**Reasons To Accept:**

- Paper is written with clarity and easy to follow.
- Extensive experiments to show the effectiveness of ReVise against previous methods and also its use in self-training.
- Method is data efficient, compared to existing approaches.
- This method also allows for few-shot self-training, further addresses the issue of scarce annotations.

**Reasons To Reject:**

- I had a hard time understanding the results because of the various metrics. While the different metrics show ReVise performs the best, it would be nice if the authors can give a better idea of what metrics to look at and how to interpret them.
- Unclear evaluation: not sure whether the evaluation is conducted on answers or explanations.

**Reproducibility:**

4: Could mostly reproduce the results, but there may be some variation because of sample variance or minor variations in their interpretation of the protocol or method.

**Reviewer Confidence:**

2: Willing to defend my evaluation, but it is fairly likely that I missed some details, didn't understand some central points, or can't be sure about the novelty of the work.

**Typos Grammar Style And Presentation Improvements:**

- Too many tables / figures (especially in page 6, 7, and 8.) Some can be moved to appendix.
- Some citations should use \citet

---

> ### Author Rebuttal · Authors · 2023-08-28
>
> We deeply appreciate your thoughtful review and the acknowledgment of our work's strengths! We've provided further clarifications to address your concerns:
>
> 1.**Evaluation Target**: The metrics we reported are evaluated on the explanations.
>
> 2.**Clarifying Metrics**:
>
> To assist in understanding the diverse metrics, we've categorized them into two primary groups:
>
> 1. Traditional metrics from "Microsoft COCO Caption Evaluation" and BertScore.
> 2. GEval: A novel metric leveraging models like GPT3.5/GPT4 and chain-of-thought reasoning
>
> Here's a succinct breakdown:
>
> |Metric  |	Description |
> |:-------------:|:-------:|
> |Bleu	|Measures the overlap of n-grams between the generated explanations and the reference explanations.|
> |Meteor	|Harmonizes precision and recall by considering synonymy, stemming, and word order in the explanations.|
> |Rouge	|Focuses on the overlap of n-grams, specifically the recall between generated and reference explanations.|
> |Cider	|Considers the consensus of terms in the generated explanations relative to multiple reference explanations.|
> |Spice	|Evaluates semantic propositions within the generated explanations and contrasts them with the references.|
> |BertScore |Measures the similarity between the generated and reference explanations using contextual embeddings from BERT.|
> |G-Eval|Evaluates the logic and reasoning of the generated explanations using GPT3.5/GPT4 and take the references as examples.|
>
> In the realm of vision-language explanation generation, no singular metric can comprehensively evaluate the quality and correctness of the generated text. Each metric we used serves a distinct purpose and examines a different aspect of the generated explanations.
>
> 3.**Iterative Explanation Quality**:
>
> We provide a table for the e-SNLI-VE dataset, We tested the quality of the samples that stops at the first step (referenced as "step1"), the quality of the samples that stops at the second step (referenced as "step2"), and the quality of the samples that stops at the third step (referenced as "step3")
>
> ​​The table illustrates the consistency in explanation quality across steps. This suggests that even for samples that require deeper iterations for correctness, the explanation quality remains commendably robust.
>
> | | B1         | B2     | B3 | B4 | M | R-L | CIDEr | S |
> |-------------|-------|-------|----------|-----|-----------|---------|-----|-----------|
> | step1                    | 36.7|23.7|15.8|10.8|18.2|31.1|104.0|31.4|
> | step2                    | 36.5|23.5|15.7|10.7|18.1|31.0|102.9|31.3|
> | step3                    | 37.7|25.8|17.6|12.2|18.7|31.6|110.9|34.7|
>
> 4.Thank you for your detailed efforts in pointing out the areas of improvement regarding typos, style, and presentation. We really appreciate it and will certainly act on them for improvement.
>
> We hope these explanations provide clarity on your concerns. Thank you again for the time and effort you dedicated to our paper's review.

---

### Official Review · Reviewer_UABU · 2023-08-05

**Soundness:** 3

**Excitement:**

4: Strong: This paper deepens the understanding of some phenomenon or lowers the barriers to an existing research direction.

**Paper Topic And Main Contributions:**

In this manuscript, the authors address the visual question answering problem, aiming to provide insightful explanations together with the answers. The proposed approach, namely the Recursive Visual Explanation algorithm (ReVisE), recursively generates an answer, together with an explanation, based on the encoded image. The generated answer and explanation will be used for the next step in the recursion. The motivation is to allow the model to improve the answer and explanation step by step in the recursion until it converges. The authors also proposed a few-shot self-training mechanism to train the model with low annotated explanation resources. Experimental results are shown to support the proposed method.


**Questions For The Authors:**

My main concern is the convergence of such recursion. First, the answer and explanation generation involves a  sampling, with which $A_n = A_{n-1}$ doesn't necessarily lead to convergence. Second, why would the model tend to converge? There is no source of "improvement" at each step. Providing clear proof might be challenging given the complexity of LLM, but it would be better to give a persuasive motivation for the possible step-wise improvement that leads to convergence.


**Reasons To Accept:**

Recursive revising the answer and explanation is a promising direction towards explanation AI.
The self-training mechanism is interesting.
The manuscript is well organized, and the writing is easy to follow.

**Reasons To Reject:**

The convergence of the algorithm is questionable.
The improvement of most metrics is somewhat marginal, given times of recursion as the price.

**Reproducibility:**

4: Could mostly reproduce the results, but there may be some variation because of sample variance or minor variations in their interpretation of the protocol or method.

**Reviewer Confidence:**

5: Positive that my evaluation is correct. I read the paper very carefully and I am very familiar with related work.

---

> ### Author Rebuttal · Authors · 2023-08-28
>
> Thank you for your detailed review and insightful feedback on our paper! We deeply appreciate the time and effort you've dedicated to understanding our work. We'd like to address your concerns:
>
> 1. **Convergence Issue**: As you rightly highlighted, a small percentage of samples do not converge. We observed this in the "Failure Case Analysis" section, where approximately 2% of samples enter an infinite loop (refer to the second case in Figure 5). To provide more insight into this issue, we analyzed the non-convergence rates for the three datasets: VQA-X (1.31%), e-SNLI-VE (1.84%), and VCR (1.67%). Investigating the root causes behind this behavior in a subset of data is indeed a compelling direction for future work.
>
> 2. **Source of Improvement**: The iterative feedback mechanism is central to our model's step-wise improvement. By taking the explanation from one iteration and using it as input for the next, the model refines its interpretation and visual attention. Conceptually, it's analogous to a person rephrasing a statement repeatedly to enhance clarity. While this isn't a formal proof of convergence, it offers a reasonable justification for our model's iterative improvement.
>
> 3. **Computational Considerations**: We recognize the computational overhead introduced by recursion. However, as evidenced in Table 6, we found that limiting the iterations to just three steps provides optimal performance. This suggests that a modest number of iterations is sufficient for our model, mitigating excessive computational costs.
>
> We hope these explanations address your concerns. We're grateful for the opportunity to clarify and improve our paper's presentation. Your feedback has been invaluable.

---

### Meta-Review · Area_Chair_DRFi · 2023-09-16

**Recommendation:** 5

**Metareview:**

This paper proposes a recursive visual explanation method for step-by-step vision-language explanation generation. The authors have done a nice job of rebuttal. In general, all the reviewers are positive (or neural) about the paper, commenting that (1) the paper is well written, (2) experiments are sufficient, and (3) the few-shot self-training mechanism is interesting. One reviewer questioned about the novelty of the method, which the AC also agrees. Overall, the AC thinks that the merits of the paper outweigh the flaws, and is positive about the paper.

---

### Decision · Program_Chairs · 2023-10-07

**Decision:**

Accept-Main

**Comment:**

This paper proposes a recursive visual explanation method for step-by-step vision-language explanation generation. The authors have done a nice job of rebuttal. In general, all the reviewers are positive (or neural) about the paper, commenting that (1) the paper is well written, (2) experiments are sufficient, and (3) the few-shot self-training mechanism is interesting. One reviewer questioned about the novelty of the method, which the AC also agrees. Overall, the AC thinks that the merits of the paper outweigh the flaws, and is positive about the paper.